

# A P2P multi-path routing algorithm based on Skyline operator for data aggregation in IoMT environments

Ismail Kertiou[1], Abdelkader Laouid[1], Benharzallah Saber[2], Mohammad Hammoudeh[3] and Muath Alshaikh[4]

[1] LIAP Laboratory, University of El Oued, El Oued, Algeria
[2] Computer Science, University of Batna 2, Batna, Algeria
[3] Information and Computer Science Department, King Fahad University of Petroleum and Minerals, Dahran, Saudi Arabia
[4] Computer Science Department, College of Computing and Informatics, Saudi Electronic University, Riyadh, Saudi Arabia

Corresponding author
Abdelkader Laouid,
abdelkader-laouid@univ-eloued.dz

## ABSTRACT

The integration of Internet of Things (IoT) technologies, particularly the Internet of Medical Things (IoMT), with wireless sensor networks (WSNs) has revolutionized the healthcare industry. However, despite the undeniable benefits of WSNs, their limited communication capabilities and network congestion have emerged as critical challenges in the context of healthcare applications. This research addresses these challenges through a dynamic and on-demand route-finding protocol called P2P-IoMT, based on LOADng for point-to-point routing in IoMT. To reduce congestion, dynamic composite routing metrics allow nodes to select the optimal parent based on the application requirements during the routing discovery phase. Nodes running the proposed routing protocol use the multi-criteria decision-making Skyline technique for parent selection. Experimental evaluation results show that P2P-IoMT protocol outperforms its best rivals in the literature in terms of residual network energy and packet delivery ratio. The network lifetime is extended by 4% while achieving a comparable packet delivery ratio and communication delay compared to LRRE. These performances are offered on top of the dynamic path selection and configurable route metrics capabilities of P2P-IoMT.

## INTRODUCTION

The past two decades witnessed a sharp increase in the number of smart devices that are equipped with sensors and actuators. These Internet-connected objects have revolutionized the healthcare landscape, improving patient care and service delivery. However, real-time communication requirements pose significant challenges, hindering seamless information exchange and jeopardizing patient outcomes (*Hammoudeh & Newman, 2015b*). As the demand for healthcare applications continues to grow, data routing protocols must

proactively address these challenges to enable efficient communication, data sharing, and medical service delivery.

The application of the Internet of Things (IoT) and wireless sensor networks (WSN) in healthcare applications has caught the attention of researchers to facilitate communication between medical practitioners and patients to reduce the costs of healthcare. The essential task of such Internet of Medical Things (IoMT) is to collect physiological measurements, *e.g.*, blood pressure and body temperature. In this context, the efficient transmission of data is a critical task. Therefore, the data routing protocols have a substantial impact on such a time-sensitive IoMT environment.

WSN represents an essential element of IoMT applications (*Hammoudeh et al., 2015*; *Alsboui et al., 2011*). These nodes have constrained processing power, memory, and battery capacity. Communication between nodes and the network gateway often occurs through more than one hop, which requires multi-hop routing protocols. Routing protocols ensure the necessary reliable connectivity and real-time communication in IoMT applications. In addition, they manage network congestion to ensure the required availability of critical medical services.

When applied to IoMT, classical routing protocols suffer from several limitations including increased latency, high power consumption, security vulnerabilities, and configuration and maintenance complexity. These routing protocols often result in high end-to-end communication delays and data loss, which may impact the quality of care and patient safety. Therefore, it is essential to employ reliable routing protocols to realise the benefits of IoMT connectivity. Implementing peer-to-peer (P2P) routing protocols in IoMT networks has the potential to increase network scalability, reduced communication latency, decentralize network control, and enhance data confidentiality and security. P2P protocols meet the distributed nature and unique demands of IoMT applications, facilitating reliable and efficient connectivity among interconnected medical devices.

To respond to the emerging routing requirements, the RoLL working team presented the IPv6 Proactive Routing Protocol (RPL) for Low-power and Lossy Networks (LLN) (*Winter et al., 2012*). Based on objective functions (OFs), RPL assembles a logical topology known as the Destination Oriented Directed Acyclic Graph (DODAG). By considering many routing metrics, network nodes select the most suitable parent from their list of immediate neighbours. A reactive Lightweight On-Demand Adhoc Distance vector routing protocol-next generation (LOADng) was proposed in *Clausen, Yi & Herberg (2017)*. LOADng works on the premise that LLNs are mostly inactive and only need to discover the route when necessary. LOADng is better suited for P2P communication IoT applications, *e.g.*, Sensing as a Service (SaaS) which can provide remote medical monitoring services and facilitate the integration of IoMT by offering a platform for collecting and analyzing medical data, thereby contributing to improving healthcare and sensor-based medical applications *Yi, Clausen & Igarashi (2013)*. These protocols provide a strong basis for exploring multi-path P2P routing protocols which is a critical aspect of our research. Multi-path routing protocols offer innovative solutions to address the challenges copied with healthcare applications, particularly reliability, energy efficiency, and robustness.

Several P2P multi-path routing protocols were developed based on RPL (*Zhao, Ho & Chong, 2016*; *Araujo et al., 2018*; *Safara et al., 2020*) and LOADng (*Sasidharan & Jacob, 2018*; *Sobral et al., 2019b*; *Adhikary et al., 2022*), but they suffer from the following limitations:

- High power consumption at node levels caused by complex computing and frequent communications.
- High communication latency and transmission delays due to the frequent search for multiple paths.
- A limited and fixed number of routing metrics are used to select the best route.

This article addresses the data routing challenge in the IoMT context, highlighting the potential benefits of utilizing routing protocols from the broader IoT domain. A P2P routing protocol, called P2P-IoMT, which is based on LOADng is presented. P2P-IoMT ensures on-demand routing over a dynamic topology to meet the quality of service (QoS) requirements of different users. The main contributions of this research are as follows:

- LOADng is extended with a composite dynamic routing metric to allow nodes to select the optimal parent during the route discovery phase. Each application selects its suitable routing metrics with optimal weightings.
- Several routing metrics suited for IoMT applications are defined. Further, the option of adding new routing metrics when a new requirement appears is provided.
- The Skyline technique is used to assist nodes in selecting the optimal parent by leveraging its strengths in multi-criteria decision-making.

The rest of the paper is organized as follows. 'Related Work' reviews the related work on routing in LNNs. 'Routing Metrics and System Description' presents the various routing metrics and gives a general description of the proposed P2P-IoMT. 'A New Technique for Best Route Selection' gives the specifications of P2P-IoMT. 'Evaluation Results and Analysis' presents P2P-IoMT's experimental performance evaluation results and analysis. Finally, 'Conclusion' concludes the paper and outlines future work directions.

## RELATED WORK

Given the strict QoS requirements of IoMT applications, P2P routing protocols can support their application scalability, real-time communications, reliable data delivery, decentralisation, and improved data confidentiality and security (*Moffat, Hammoudeh & Hegarty, 2017*). RPL was proposed as a standard routing protocol for LLN networks (*Winter et al., 2012*). RPL's goal is to make it possible to build a DODAG, where the root of the logical topology is the gateway node. The construction of the DODAG focuses on objective functions, which makes it possible to choose the best paths in the network by considering several routing metrics such as the number of hops, latency, delivery rate, node residual energy, and link quality. An objective function allows nodes to choose their preferred parent from their immediate neighbours to reach the DODAG root. RPL accepts three different traffic patterns, namely, multipoint-to-point (MP2P), point-to-multipoint (P2MP) and P2P (*Sobral et al., 2019a*).

Research efforts in the literature considered how to enhance the P2P-RPL. To send P2P messages, each source node must initiate a route discovery process, which generates a temporary routing tree. The root of the temporary DODAG (the source node) broadcasts a P2P route discovery message, called P2P-RDO. After receiving a P2P-RDO, each node must verify the message's destination, decide on joining the DODAG, select its temporary preferred parent node, and forward the P2P-RDO. Although P2P-RPL offers several advantages over traditional RPL, it also suffers from increased power consumption and high network overhead.

The GOAFR algorithm is another RPL-based routing protocol that supports P2P traffic and reduces the number of control messages (*Barriquello, Denardin & Campos, 2015*). Based on the list of DODAG roots, the GeoRank algorithm determines the shortest distance between the source node and its destination. The root with the smallest absolute angle is chosen as the mediator node for message transfer. GeoRank is vulnerable to localization errors and increased power consumption.

*Zhao, Ho & Chong (2016)* proposed the energy-efficient region-based routing protocol (ER-RPL) to reduce the overload and the energy consumption generated by the network flood when establishing the P2P routes. ER-RPL divides the network into regions and selects the P2P route based on the nodes' locations. Although the ER-RPL algorithm has significant advantages, it is complex to implement, incurs increased signalling overhead, and is not adaptable to dynamic changes in the environment.

As part of the route selection phase, several routing protocols for LLNs consider QoS and energy consumption of nodes (*Hammoudeh & Newman, 2015a*). To improve the reliability of data transmission, *Ancillotti, Bruno & Conti (2014)* proposed a cross-layer implementation of RPL called RPLca. Two libraries were used to develop RPLca. A quality estimation approach is provided in the first, and a neighbour management approach in the second. RPLca increases implementation overhead and incurs to higher energy consumption. An objective function called Quality of Service RPL (QoS-RPL) was introduced in *Mohamed & Mohamed (2015)*. QoS-RPL is based on an Ant Colony optimisation algorithm and uses the residual energy and transmission delay as a routing metric to find the best parent. In QoS-RPL, there is a trade-off between QoS and energy consumption. Also, configuring the QoS parameters in QoS-RPL is known to be a complex task.

To extend the network lifetime, *Iova, Theoleyre & Noel (2015)* proposed an RPL-based protocol to balance power consumption among nodes. This approach employs a mechanism for measuring the life of nodes and uses multiple paths to avoid depleting the energy of nodes in common locations. A routing protocol that provides reliable data transmission for IoT applications was introduced in *Qiu et al. (2016)*. This protocol used two mechanisms, the first arranges the nodes of a candidate path based on the overall delay estimation, and the second is to discover the next node for message transmission. To meet these requirements, *Araujo et al. (2018)* developed an approach to discover IoT routes using fuzzy logic. Four OFs were implemented in the LLN's networks RPL routing protocol. These OFs are automatically selected based on the application context. A new method based on RPL, called PriNergy, was proposed in *Safara et al. (2020)*. The proposed

method minimizes the power consumption of IoT devices and can significantly reduce traffic and delays. PriNergy uses QoS of IoT applications, where TDMA time slot is applied to synchronize between a sender and a receiver to reduce power consumption. However, PriNergy inherits some of the limitations from RPL and QoS-based protocols, including increased complexity, power consumption, signalling overhead, and difficulties adapting to dynamic environmental changes.

Unlike the RPL proactive method, the next generation lightweight Advanced On-demand Ad hoc Distance Vector routing (AODV), like LOADng, are reactive protocols. The main concept of LOADng is that LLN is inactive and only generates a path when a node has data to send (*Clausen, Yi & Herberg, 2017*). All control messages in AODV are exploited in the LOADng route discovery process (*Perkins, Belding-Royer & Das, 2003*). The control messages used in LOADng are *Clausen & de Verdiere (2011)*:

- Route Request (RREQ): Discovers all other nodes in the network.
- Route Reply (RREP): Created by the destination of an RREQ in response to routing requests.
- Route Reply Acknowledgement (RREP-ACK): Used to reply to the sender of the received RREP.
- Routing Error (RERR): Used to send a notification about routing problems during data transmission.

Even though the RPL protocol is widely adopted for IoT communications, several research papers study its limits and disadvantages (*Anusha & Pushpalatha, 2023*; *Sobral et al., 2019b*; *Sousa et al., 2017*). Among these limitations are the limited support for P2P traffic and the weak adaptation to the dynamic changes of the various routing metrics. These serious limitations make RPL unsuitable to support additional communication demands imposed by the IoMT applications. In *Yi, Clausen & Igarashi (2013)*, a comparison to evaluate the performance of the LOADng and RPL protocols in different traffic models was presented. The presented results show that RPL provides better outcomes for MP2P traffic. However, the results obtained for P2MP and P2P are better with LOADng traffic. Therefore, RPL is better suited for data collection applications, whereas LOADng is more suited for coordination with generalized traffic. Due to the operational mode of the generated request, LOADng takes more time to discover the route. In IoT applications, P2P results in significant traffic increase, especially in the SaaS setting, because any gateway must every time find the best way to reach the relevant nodes (*Sobral et al., 2019a*).

To solve the problem of asymmetric links, *Perkins et al. (2022)* proposed an AODV-RPL where a node initiates the route discovery process to a destination when the current route does not meet an application requirement. The source node sends a DIO-RREQ message to create an RREQ instance to find a path to the destination node. The field $S$ in DIO-RREQ indicates whether a route is symmetrical or asymmetrical. Upon receiving a DIO-RREQ, each node must verify whether the route is symmetric or asymmetric, update $S$, and join the RREQ instance. Nodes must decide how to create routes based on DIO-RREQ when it arrives at their destination. However, AODV-RPL is still an IETF Internet Draft and may change until it is fully defined.

*Hossain, Sreenan & Alberola (2016)* proposed a Neighbor Disjoint Multipath scheme, named LOADng+NDM. LOADing+NDM first discovers the main path, which is generally the shortest between the source and the destination. Then it builds other backup routes. LOADng+NDM does not consider power consumption or link quality information in route selection. *Sasidharan & Jacob (2018)* defined a new mixed routing metric, LRRE, for LOADng to reduce network congestion and extend the node life. Three routing metrics are employed to select the best route between a source node and destination: hop count, node residual energy, and the total number of live routes. LRRE uses a limited number of metrics; moreover, an incorrect adjustment of the parameters of the proposed routing metric can decrease the network's performance.

*Sobral et al. (2019b)* developed an extension for LOADng, called LOADng-IoT, to help nodes find a route to a gateway. A node broadcasts an RREQ-IoT as a standard RREQ from default LOADng. LOADng searches for an IoT gateway by broadcasting RREQ-IoT. This RREQ-IoT does not specify a destination and can be answered by any gateway. The gateway creates an RREP-IoT message and forwards it to the requesting node when it receives the RREQ-IoT message. LOADng-IoT requires inserting an extra field on the default LOADng control messages, and the route cache mechanism can increase memory usage.

The study by *Diniesh et al. (2022)* aims to select the optimal fitness functions using routing metrics for wireless body area networks (WBAN). To evaluate routing performance in WBAN, the two evolutionary algorithms particle swarm optimization (PSO) and teaching learning-based optimization (TLBO) are used. *Adhikary et al. (2022)* developed a topology with an optimized number of relay nodes and an efficient routing algorithm. All sensor nodes are connected to at least one relay node. This topology ensures minimum hops between the body sensors and the destination node. Additionally, multi-casting is used to reduce the unnecessary transmission of packets. The last two approaches were designed specifically for WBAN and suffer from several problems, including increased energy consumption, high latency and delay, and low reliability and fault tolerance.

A framework to improve the QoS in IoMT applications was proposed in *Sobral et al. (2018)*. Two fuzzy systems were developed to enhance routing protocols' performance and deliver QoS with energy preservation. The first system makes the reading process of RFID tags faster and more reliable. The second system uses four routing metrics (node energy, number of hops, tags' density, and link quality indicator) to select the best path. The approach is based on fuzzy systems, which can be complex when applied to IoT devices.

Reactive routing protocols offer IoMT several advantages, including conserving energy by triggering route discovery only when necessary and optimising bandwidth utilization by establishing on-demand routes to reduce communication overload. Moreover, they adapt to network topology changes, which is crucial in IoMT environments where devices can be mobile. Reactive routing protocols are well-suited for large-scale networks and minimize overload by avoiding transmitting unnecessary routing information. However, the protocol choice depends on application specifications and network constraints, considering performance requirements, energy limitations, and deployment characteristics.

Below is a list of the common limitations of routing protocols for IoT and IoMT networks.

- P2P communication is not supported in many IoT routing protocols.
- Many protocols incur complex calculations and frequent communications which leads to an increase in power consumption.
- Searching for multiple paths may cause increased latency and transmission delays.
- A limited and fixed number of routing metrics are considered when selecting the best route.
- Several metrics specific to IoT and IoMT are not considered when choosing the best routes.
- Inefficient traffic management.

Considering the limitations above, we propose a new P2P routing protocol, called P2P-IoMT, that allows routes to be selected dynamically and on demand. P2P-IoMT is built on LOADng with a new objective function to choose the best parent during route discovery. Depending on the application needs defined by the requested routing metrics, the objective function compares and selects the optimum route from the source to the destination using the Skyline operator and the Euclidean distance. Furthermore, the proposed routing protocol considers many routing metrics to suit the diverse needs of IoMT applications.

## ROUTING METRICS AND SYSTEM DESCRIPTION

IoMT applications possess distinct characteristics that make common IoT routing protocols not suitable for their applications. In the following, we present the key design objectives of the proposed P2P-IoMT routing protocol for IoMT.

- Latency management: Real-time data transmission is a crucial requirement in IoMT. In such environments, the routing protocols should be designed to minimize latency and ensure rapid delivery of messages, which is vital in real-time medical monitoring applications. P2P-IoMT uses the delay, link quality, and availability metrics to choose the best paths with the lowest possible delay.
- Transmission reliability: The loss or modification of medical data could have severe consequences. P2P-IoMT routing protocols prioritize data transmission reliability by employing mechanisms like error control, packet re-transmission, and congestion detection to ensure data integrity. Using the link quality and availability metrics allows the selection of paths with the highest reliability level.
- Security and confidentiality: Medical data necessitates utmost confidentiality and protection against unauthorized access. P2P-IoMT allows the selection of paths based on a 'security level' metric.
- Energy consumption management: IoMT devices often operate on limited energy sources. Hence, IoMT routing protocols should be optimised to minimize energy consumption in connected devices through effective energy management techniques, *e.g.*, deep sleep and selective wake-up. P2P-IoMT uses a P2P routing which conserves energy through direct communication between nodes, reducing communication overhead, and energy balancing among nodes.

In P2P-IoMT, the routing path selection procedure utilises many routing metrics to adapt to the various application requirements. A new routing metric can be added to the protocol if new requirements emerge. P2P-IoMT extends the author's earlier work (*Laouid et al., 2017*). *Laouid et al. (2017)* considers the energy and the hop count as the only metrics in calculating the best route. This study introduces several other routing metrics to adapt to the IoMT application requirements.

The following subsection presents the various routing metrics employed during the routing path selection step. Then, the system model and P2P-IoMT specifications are given.

## Routing metrics

An objective function optimizes a specific metric for finding a routing path to meet a particular application requirement or reduce the communication cost. Each metric represents some context information used as a criterion for selecting the best route. The objective function can combine two or more routing metrics.

An IoMT application requirement may ultimately change. Hence, a fixed routing metric can not meet the varying application needs. The combination of the expected number of re-transmissions (ETX) and residual energy measurements, for example, allows the construction of more reliable communication pathways while extending the network lifetime. Such pathways may result in an inconsistent access time, which is unsuitable for real-time applications. As a result, different criteria must be combined to determine the best path.

In this section, We define a set of routing metrics corresponding to various routing QoS requirements.

- **Hop count:** This metric measures how many hops a message took to travel from a source node $s$ to a destination node $d$. This metric allows the selection of the shortest bath from $s$ to $d$. The hop count (Hc) function determines the number of nodes on a given route $R$ using Eq. (1).

$$Hc(R(s,d)) = |(n_1,..,n_k)| \qquad (1)$$

- **Energy:** This metric refers to the node's residual battery level. With this metric, it is feasible to avoid selecting low-energy routes allowing the network lifetime to be extended. The energy (En) function, defined in our previous work *Laouid et al. (2017)*, estimates the energy factor value of a given route using Eq. (2).

$$En(R(s,d)) = w_1 \times \mu(R(s,d)) + w_2 \times \sigma(R(s,d)) \qquad (2)$$

where $\mu(R(s,d))$ represents the energy mean of the route $R(s,d)$, $\sigma(R(s,d))$ represents the energy standard deviation of the route $R(s,d)$ where $w_1 + w_2 = 1 \ \forall \ w_1 \geq 0$ and $w_2 \geq 0$.

- **Delay:** This metric measures the time it takes a message to travel from $s$ to $d$. For applications requiring real-time message delivery guarantees, this measure can be used to choose the route with the least delay. The delay (Dl) function defined in Eq. (3),

calculates the sum of the transfer time of all the links in the route $R$.

$$Dl(R(s,d)) = \sum_{i=0}^{k-1} dl(n_i, n_{i+1}) \tag{3}$$

where $dl(n_i, n_{i+1})$ represents the time to carry a message from the node $n_i$ to $n_{i+1}$.

- **Service cost:** It is crucial to compute the service cost for a message to travel from $s$ to $d$ in an IoMT network and with the introduction of SaaS (*Hammoudeh et al., 2020*). The cost (Co) function, defined in Eq. (4), calculates the sum of each node's service cost in the route $R$.

$$Co(R(s,d)) = \sum_{i=1}^{k-1} sc(n_i) \tag{4}$$

where $sc(n_i)$ represents the service cost necessary to use node $n_i$ as a bridge.

- **Link Quality:** This metric is required for applications that need a high level of communication reliability. Several parameters, including the ETX, link quality Level (LQL), and received signal strength (RSS), can be used to determine the link's quality. Eq. (5) defines the link quality (Lq) function, which calculates the minimum between the link qualities of each link in the route $R$. The "weakest link" approach is applied to calculate Lq (*Fang et al., 2008*; *Chaudhary & Raghav, 2016*).

$$Lq(R(s,d)) = min_{i=0}^{k-1} lq(n_i, n_{i+1}) \tag{5}$$

where $lq(n_i, n_{i+1})$ represents the link quality between node $n_i$ and $n_{i+1}$.

- **Security level:** This metric addresses security-sensitive applications. The security level (Sl) function estimates the minimum between the security levels of each link in the route $R$, as represented in Eq. (6) (*Fang et al., 2008*; *Chaudhary & Raghav, 2016*)

$$Sl(R(s,d)) = min_{i=0}^{k-1} sl(n_i, n_{i+1}) \tag{6}$$

where $sl(n_i, n_{i+1})$ represents the security level between node $n_i$ and $n_{i+1}$.

- **Availability level:** This metric is relevant for mission-critical applications. Eq. (7) defines the Availability Level (AL) function, which estimates the minimum between the availability levels of each link in the route $R$ (*Fang et al., 2008*; *Chaudhary & Raghav, 2016*).

$$Al(R(s,d)) = min_{i=0}^{k-1} al(n_i, n_{i+1}) \tag{7}$$

where $al(n_i, n_{i+1})$ represents the availability level between node $n_i$ and $n_{i+1}$.

One or more metrics can be utilized to determine the best path depending on the unique needs of each IoMT application. P2p-IoMT is designed to accept other routing metrics to meet the unique needs of emerging applications.

## P2P-IoMT specifications

This section gives the technical specifications of the proposed P2P-IoMT routing protocol. Routing metrics listed in 'Skyline Operator' are used to calculate the best route for a particular IoMT application.

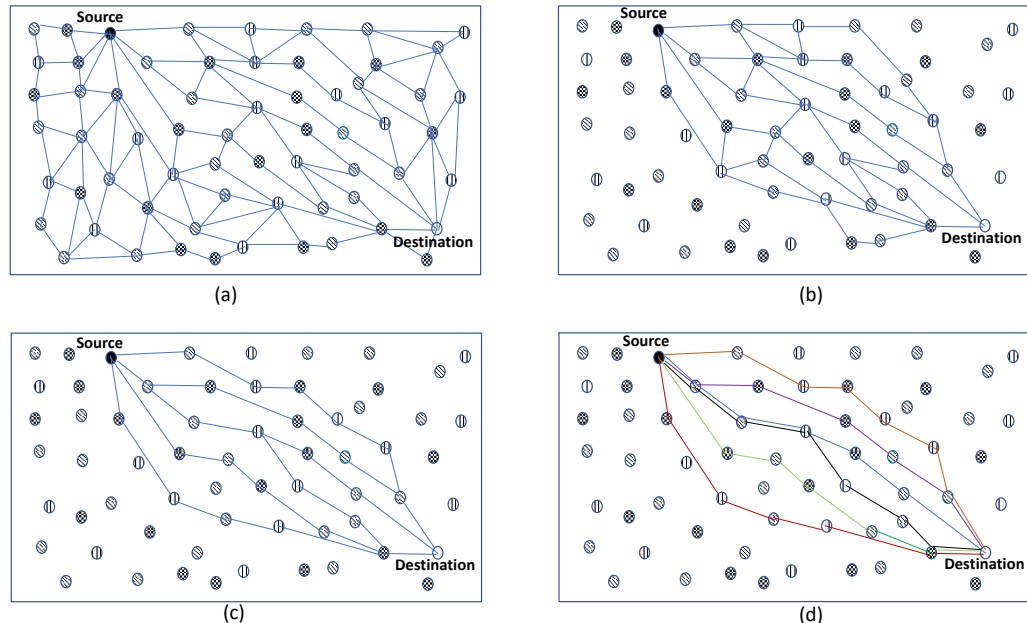

**Figure 1** (A–D) The selection process of the best route.

The network is represented as a graph $G$ that is not necessarily fully connected and supports multi-hop communications. As shown in Fig. 1, $G$ consists of a set $N$ of $n$ nodes, each representing a device, and a set $L$ of links connecting these devices. We define $R(s, d)$ as a sequence of nodes $(n_0, n_1, .., n_k)$ representing the path from $s$ to $d$ while ensuring that the following requirements were met:

- $n_0 = s$
- $n_k = d$
- $n_i \neq n_j$ for $i \neq j$
- $(n_i, n_{i+1}) \in L$ for $0 \leqslant i \leqslant k - 1$

To choose the optimal route, the source node first broadcasts a route request message (P2P-RReq) to all of its neighbours. The content of the P2P-RReq is described in Table 1 and summarised as follows: (1) P2P-RReq message type, (2) destination IP Address, (3) hop count between $n_i$ and the source node, (4) number of metrics that will be sent, (5) cumulative value of each necessary routing metric of the previous optimal route, (6) weighting value of each critical routing metric, and (7) hop limit. After receiving a P2P-RReq, a node increases the hop count value and computes and updates the cumulative difference value of each metric between the current parameter of the node and its predecessors. Finally, every node determines using an objective function which parent is the best for forwarding messages to the gateway by utilising the saved hop count value and the cumulative metrics.

The P2P-RReq processing method ensures all broadcast messages follow the shortest path to the source node. The distributed computation of the path minimises energy

| Table 1 | The components of the route request message P2P-RReq. | | | | | | | | | | | |
|---|---|---|---|---|---|---|---|---|---|---|---|---|
| head | P2P-RReq | Dest Adr | Hop | Num Met | $RM_1$ | $W_1$ | $RM_2$ | $W_2$ | ... | $RM_n$ | $W_n$ | hop limit |

consumption and the number of transmitted messages. P2P-IoMT proceeds with the following steps:

1. **Broadcast and distributed computation**: A node must perform some computation before broadcasting its P2P-RReq, such as computing the cumulative value of each needed routing metric. Table 2 shows how the best routes are kept in the routing table; the table is arranged into classes, with each class containing routes with the same number of hops. Each node broadcasts the best cumulative value of the used routing metrics just once after a predetermined time from receiving the first P2P-RReq. A node can keep other routes as backups or alternative routes. This approach offers a solution for the joint route problem and the reduction of message exchange during the network initialisation phase. A predetermined delay time is imposed to receive all anticipated P2P-RReq messages before propagating the best route. Although each node incurs an overhead because of the preset delay, this method reduces the joint route problem. The purpose of a network flood is to ensure that the routing table of each node has several entries arranged by hop count to reach the destination. When the optimal path required routing metrics of the objective function are below a specified threshold, the node requests the source node to initiate a discovery process to find alternative paths. Once a message has reached its hop limit, it can not be transmitted further. As a result of reducing node access, resource utilisation at the node level is reduced. P2P-IoMT does not save a routing table for all nodes in the network because it operates in a reactive mode. This strategy reduces routing overhead and memory usage. In addition, P2P-RReq only contains the application request's needed metrics, which reduces the message size.

2. **Route selection**: The objective is to determine the optimal route between *s* and *d* according to the application's QoS needs. The application's QoS need is defined as a request, representing the weightings of various routing metrics of the required objective function. The selection process is divided into three parts. Just the first classes with the fewest hops in the first step are examined to reduce the selection field, as shown in Fig. 1B. In the second step, the Skyline operator chooses the routes that best satisfy the application request, as illustrated in Fig. 1C. The Euclidean distance is then applied in the third stage to rank the best routes to overcome the Skyline problem, not showing the different routes in priority order, as seen in Fig. 1D. Increasing the number of routes in the selected set allows the identification of better routes with higher objective function values. Other classes with a high number of hops can be used to achieve such an increase.

Below is a description of other P2P-IoMT message types:

1. Route Reply (P2P-RRep): When the destination receives the P2P-RReq, it must respond with a P2P-RRep. The P2P-RRep is returned *via* the preferred parent specified during P2P-RReq forwarding.

**Table 2  Node routing table at time $t$.**

| | Node $n_i$ | | | | | | | |
|---|---|---|---|---|---|---|---|---|
| | source | Neighbor-node | Hops | $RM_1$ | $RM_2$ | $RM_3$ | ... | $RM_n$ |
| Class 0 | sink | $n_{i1}$ | $h_0$ | $RM1_{i1}$ | $RM2_{i1}$ | $RM3_{i1}$ | ... | $RMn_{i1}$ |
| | sink | $n_{i2}$ | $h_0$ | $RM1_{i2}$ | $RM2_{i2}$ | $RM3_{i2}$ | ... | $RMn_{i2}$ |
| | sink | $n_{i3}$ | $h_0$ | $RM1_{i3}$ | $RM2_{i3}$ | $RM3_{i3}$ | ... | $RMn_{i3}$ |
| Class 1 | sink | $n_{i4}$ | $h_1$ | $RM1_{i4}$ | $RM2_{i4}$ | $RM3_{i4}$ | ... | $RMn_{i4}$ |
| | sink | $n_{i5}$ | $h_1$ | $RM1_{i5}$ | $RM2_{i5}$ | $RM3_{i5}$ | ... | $RMn_{i2}$ |
| | sink | $n_{i6}$ | $h_1$ | $RM1_{i6}$ | $RM2_{i6}$ | $RM3_{i6}$ | ... | $RMn_{i6}$ |
| | . | . | . | . | . | . | ... | . |
| Class m | . | . | . | . | . | . | ... | . |
| | sink | $n_{ik}$ | $h_m$ | $RM1_{ik}$ | $RM2_{ik}$ | $RM3_{ik}$ | ... | $RMn_{ik}$ |

2. Route Reply Acknowledgement (P2P-Rrep-ACK): This message is used to confirm the receipt of a routing response message (P2P-RRep).
3. Route Error (P2P-Rerr): This message reports routing errors. It is sent when nodes detect connectivity or route availability issues. Nodes that receive a P2P-Rerr update their routing table accordingly.

# A NEW TECHNIQUE FOR BEST ROUTE SELECTION

This section proposes a technique for selecting the best relevant route based on the application needs as defined in the selected routing metrics. The Skyline approach is discussed. Then, the different phases of the proposed protocol are explained. Finally, a clarifying example is provided to demonstrate the proposed technique.

## Skyline operator

The Skyline operator (*Borzsony, Kossmann & Stocker, 2001*) and its variations, such as dynamic Skyline (*Papadias et al., 2003*) and reverse Skyline (*Dellis & Seeger, 2007*), recently attracted research interest in multiple criteria decisions making. In this subsection, we present the skyline query and describe how to utilize it to solve the route selection problem.

A typical example of the Skyline application is to select the hotel with the lowest price and the closest proximity to the beach (*Borzsony, Kossmann & Stocker, 2001*). Hotel $h1$ with a price of 600 and a distance of two miles from the beach is preferable to hotel $h2$ with a price of 700 and a distance of three miles from the beach. We say that $h1$ dominates $h2$.

Given a set $R$ of data points, the routes in P2P-IoMT are represented in d-dimensional space, with each dimension representing a routing metric of the routes described with correctly ordered values. In each metric, we assume that the lowest value is preferred. For routes $R_i$ and $R_j$, route $R_i$ is better than the route $R_j$ with respect to $R$, if $R_i$ is not greater than $R_j$ in all metrics. Furthermore, $R_i$ must be smaller than $R_j$ in at least one dimension. We say that $R_i$ dominates $R_j$. Formally, a route $R_i$ dominates $R_j$, is denoted as $R_i < R_j$, if and only if:

$$R_i(k) \leq R_j(k) \tag{8}$$

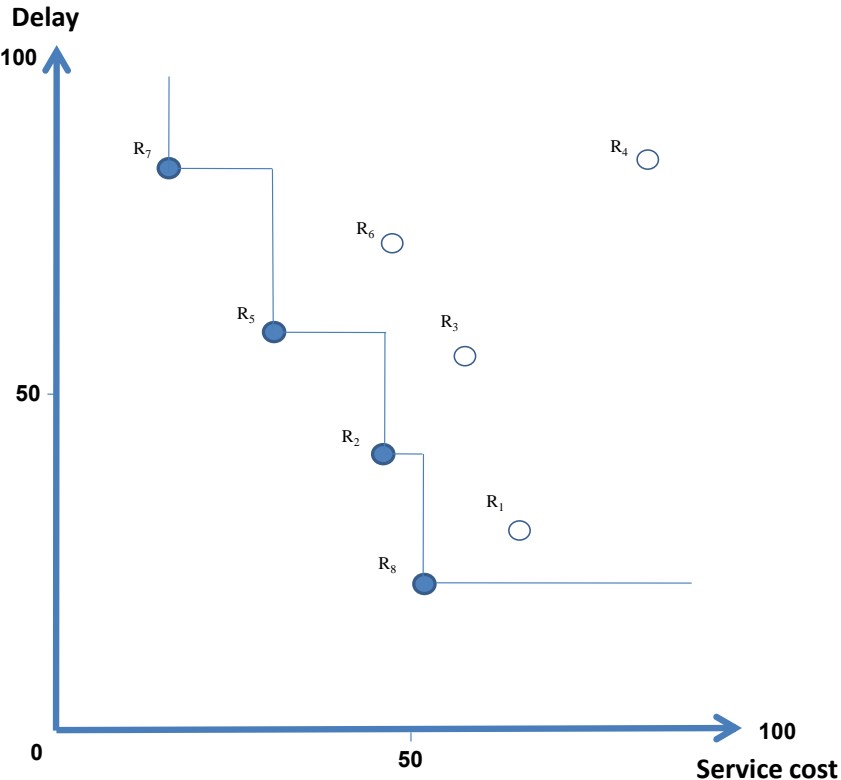

**Figure 2** Skyline routes arrangement.

$\forall\, k$ with $1 \le k \le d$ and exists $k$ with $1 \le k \le d$ such that

$$R_i(k) < R_j(k) \tag{9}$$

where $R_i(k)$ represents the value of the $k$-th routing metrics of the route $R_i$.

Figure 2 depicts an example using a two-dimensional routing metric space $R$ with the multiple routes depicted in Table 3. The service cost and latency routing metrics correspond to each route. Meanwhile, note that route $R_5$ is a better choice than $R_6$, since $R_5$ has lower service cost and delay values than $R_6$, *i.e.*, $R_5$ dominates $R_6$. Based on this method, we can discover Skyline routes not dominated by other routes. A route selection application can only evaluate the routes in the Skyline $\{R_2, R_5, R_7, R_8\}$.

## Selection of the best route

The goal is to allow applications to select the path that best fits their QoS needs. P2P-IoMT assumes that at time $t_0 = 0$, each node has its identifications and the values of each routing metric. The hop counter and the timer reduce redundant message transmissions. Each node that receives P2P-RReq performs some calculations before broadcasting it to all its neighbours. Figure 3 shows a flowchart that explains how P2P-IoMT works. The flowchart has three levels: (1) The source node, which creates and broadcasts a route request message P2P-RReq. (2) The intermediate node receives the messages, chooses the best parent, and

**Table 3  A set of routes with two routing metrics.**

| ID routes | Service cost | Delay |
|---|---|---|
| $R_1$ | 66 | 28 |
| $R_2$ | 47 | 40 |
| $R_3$ | 59 | 55 |
| $R_4$ | 85 | 82 |
| $R_5$ | 32 | 58 |
| $R_6$ | 49 | 71 |
| $R_7$ | 18 | 81 |
| $R_8$ | 53 | 21 |

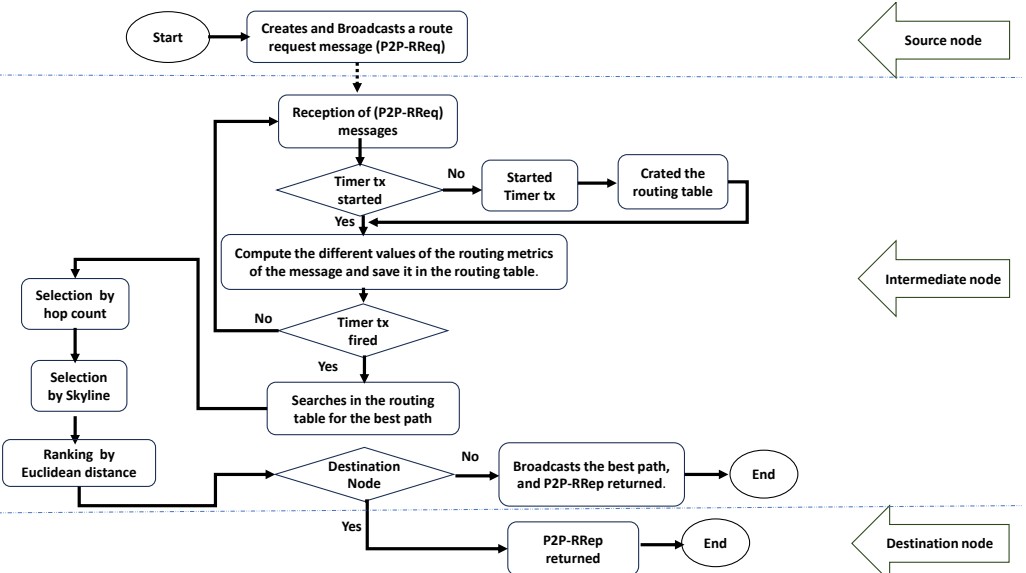

**Figure 3  A flowchart of the proposed routing protocol.**

rebroadcasts the best-found path. (3) The destination node returns the P2P-RRep message. The computation performed by a given intermediate node $n_t$ is summarised as follows:

1. Increase the hop value of the receiving P2P-RReq and, if not already started, start the timer $t_x$.
2. Compute the different routing metric values as described in the Equations $1 - 7$ to determine the different factor values of the sender node.
3. The routes with the same hops are grouped by class and saved in the routing table.
4. The timer waits $t_x$ to receive all probable P2P-RReq.
5. Finally, once the timer is expired, the node calculates the best route before broadcasting it.

At the end of the timer $t_x$, each node searches in the routing table for the optimal path to reply to the query. The route selection method is divided into three stages, which were inspired by the critical phases developed in our earlier work (*Kertiou et al., 2018*):

- **Phase 1:** The number of nodes in a route directly impacts its quality. The energy usage, response time, and service cost increase when the number of nodes increases. P2P-IoMT evaluates the path with the fewest hops while determining the best route. Then, P2P-IoMT employs the first class of routes, as shown in Fig. 1B. Increasing the class number in the selection phase of the best routes enhances the scope of discovering routes. However, this increases the calculation at the node level and reduces the node's life.
- **Phase 2:** The Skyline query reduces the search space and improves route discovery's efficiency. The skyline is the collection of all routes not dominated by one another. We only consider Skyline routes throughout the route selection process, where Skyline routes dominate non-Skyline routes due to higher routing metrics.
- **phase 3:** The objective is to find the optimal path that fulfils an application's requirements. The skyline approach allows a node to choose the best routes but does not rank them. Hence, the skyline must be complemented with a multi-criteria decision technique for this objective. As a result, the list of routes not dominated is utilized in the ranking phase. The following is the order of the routes:

    - The list of Skylines routes is considered as a matrix analysis of $Q = (q_{ij})$, $i = 1..n$, $j = 1..m$, where each line represents a route, and each dimension (column) represents a routing metric, *e.g.*, delay, energy or service cost. $N$ is the number of routes, and $M$ is the number of routing metrics. Each element of the matrix $q_{ij}$ represents the value of the routing metric $j$ of the route $i$.
    - Then use the following formula is applied to normalize the analysis matrix over $[0, 1]$:

    $$q'_{ij} = \frac{q_{i_j} - q_j^{min}}{q_j^{max} - q_j^{min}}$$

    where $q_j^{min}$ is the minimal and $q_j^{max}$ the maximal value of column $j$.
    - The Euclidean distance between each route of the matrix and the origin of the space $O$ (ideal route) is calculated as follows:

    $$d(R, O) = \sqrt{\sum_{j=1}^{m} w_j (R^j)^2}$$

    where $w_j$ represents the weight of the $j$th routing metrics required.
    - Finally, the routes are ranked in increasing order based on the Euclidean distance before choosing the first route as the optimal route.

Algorithm 1 details the procedures for selecting the best parent for each node, the acronyms are listed in Table 4.

## Application use case

We track the execution of the different P2P-IoMT steps on a use-case to demonstrate its execution in practice. Assume there is a network with 10 nodes, shown in Fig. 4, where the values of the nodes indicate the node's energy level and service cost, and the values of the links represent the latency, link quality, security level, and availability level of the link. The

**Table 4  List of abbreviations.**

| Abbreviations | Definition |
|---|---|
| *NewM* | Message received by the node |
| *BestP* | The best parent |
| *SecondP* | The parent's second choice |
| *LoclMetP* | Different values of the local metrics of the node |
| *EucDis* | Euclidean distance calculator function |

---

**Algorithm 1** Best parent selection at nodes level.

1: Input: *NewM*, *BestP*, *SecondP*; *LoclMetP*
2: Output: *BestP* and *SecondP*;
3: MAJ of the *NewM*;
4: **if** ((*NewM*) dominates (*BestP*)) or (EucDis(*NewM*) < EucDis(*BestP*)) **then**
5:     *SecondP*:= *BestP*;
6:     *BestP*:= *NewM*;
7: **else**
8:     **if** ((*NewM*) dominates (*SecondP*)) or (EucDis(*NewM*) < EucDis(*SecondP*)) **then**
9:         *SecondP*:= *NewM*;
10:     **end if**
11: **end if**

---

objective is to select the best route from node 1 to node 10 according to the application's requirements. We assume that an application requests the optimal route with the lowest latency and service cost, with weights of 0.6 and 0.4, respectively.

First, node 1 prepares the route request message P2P-RReq as needed in the application before broadcasting it to all its neighboring nodes. After receiving all potential P2P-RReq messages before the timer expired, each node generates the routing table containing the different parents connecting it to the destination node (node 10) with the required routing metrics' cumulative values. The following steps are applied to select the best route that meets the demands of the application:

- **Phase 1:** Evaluate the first three classes of routes to reduce the selection field for the best route. Then, investigate routes with a minimum, minimum plus one, and minimum plus two hops to route data from the source node (node 1) to the destination node (node 10) with the cumulative values of delay and service cost as shown in Table 5 in columns (2; 3; 4).
- **Phase 2:** Calculate the list of Skyline routes as presented in the fifth column of Table 5.
- **Phase 3:** Use the Euclidean distance to rank the list of Skyline routes as presented in the sixth column of Table 5.

Finally, as shown in the two last columns of Table 5, each node chooses the best parent, *i.e.,* the node which provides the best path to the destination node, to transmit back to their neighboring nodes and saves the remaining parents as secondary parents.

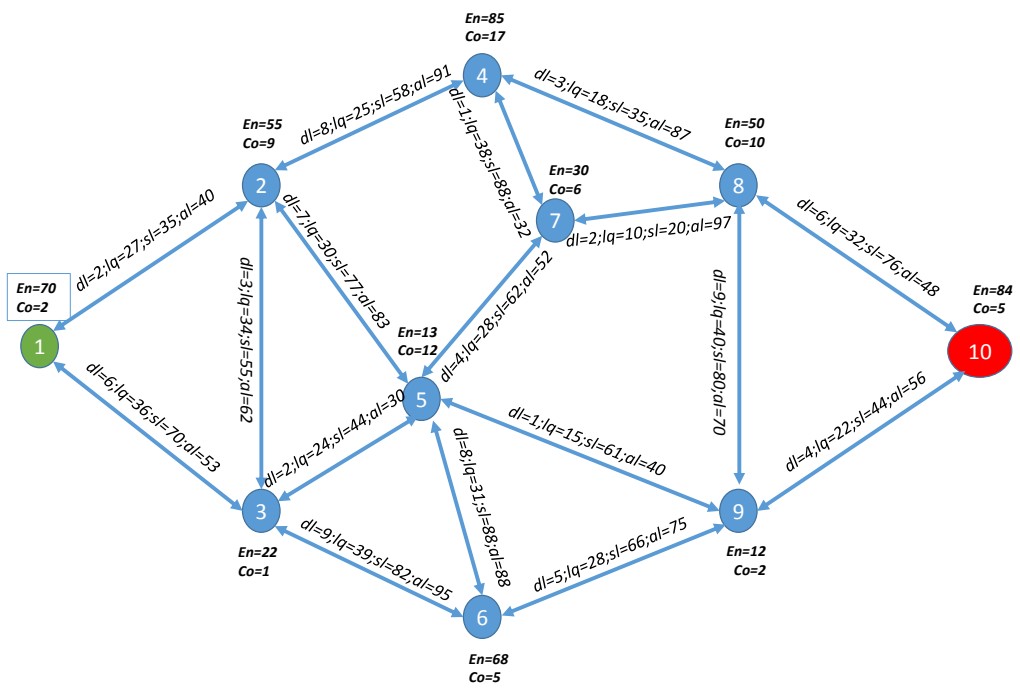

**Figure 4   A use case example network with 10 nodes.**

# EVALUATION RESULTS AND ANALYSIS

To prove the efficiency of P2P-IoMT, we conduct experiments that focus on the selection of the best route while meeting the IoT application requirements. Then, we analyze the obtained results and compare them to the conventional LOADng (with hop count as a routing measure) and LRRE (*Sasidharan & Jacob, 2018*).

## Simulation model

We adopt the same testing setup and data described in *Laouid et al. (2017)*. The experiment was run in the TinyOS simulator (TOSSIM) to mimic and extract the TelosB sensor findings. We construct a network with 50 randomly distributed nodes throughout a 60 × 60 m$^2$ space. To calculate the communication routes, we employ the P2P-RReq broadcast discovery request. Nodes are homogeneous in that they all have the same specifications.

On Mac 802.15.4, the transmission/reception rate is 250 kbps, with a maximum message size of 29 bytes. A unique ID identifies each device. The TelosB Motes (*Prayati et al., 2010*) are used to calculate the power consumption characteristics. Each node is powered by a battery with an initial capacity of 9580*J*. The total simulation phase lasted 27.3 min. The following routing metrics were used to choose the optimal route: hop count, node energy, node cost, and transmission latency (delay).

**Table 5  Routing tables for select the best route between node 1 and node 10.**

| Nodes | Neighbors | Accumulated delay | Accumulated service cost | Best routes with Skyline | Ranking routes with Euclidean distance | Best parent | Second parent |
|---|---|---|---|---|---|---|---|
| 2 | 1 | 2 | 11 | 1–2 | 1–2 | 1 | 2 |
|   | 3 | 9 | 12 |   |   |   | 5 |
|   | 4 | 18 | 37 |   |   |   |   |
|   | 5 | 15 | 24 |   |   |   |   |
| 3 | 1 | 6 | 3 | 1–3 | 1–3 | 1 | 2 |
|   | 2 | 5 | 12 | 2-3 | 2-3 |   | 5 |
|   | 5 | 9 | 24 |   |   |   |   |
|   | 6 | 24 | 9 |   |   |   |   |
| 4 | 2 | 10 | 28 | 2–4 | 2–4 | 2 | 7 |
|   | 7 | 13 | 38 |   |   |   |   |
|   | 8 | 17 | 48 |   |   |   |   |
| 5 | 2 | 9 | 23 | 3–5 | 3–5 | 3 | 2 |
|   | 3 | 8 | 15 |   |   |   | 5 |
|   | 6 | 23 | 20 |   |   |   |   |
|   | 7 | 16 | 33 |   |   |   |   |
|   | 9 | 10 | 29 |   |   |   |   |
| 6 | 3 | 15 | 8 | 3–6 | 3–6 | 3 | 9 |
|   | 5 | 16 | 20 | 9-6 | 9-6 |   | 5 |
|   | 9 | 14 | 22 |   |   |   |   |
| 7 | 4 | 11 | 34 | 4–7 | 5–7 | 5 | 4 |
|   | 5 | 12 | 21 | 5-7 | 4-7 |   |   |
|   | 8 | 16 | 37 |   |   |   |   |
| 8 | 4 | 13 | 38 | 4–8 | 7–8 | 7 | 9 |
|   | 7 | 14 | 31 | 7–8 | 9–8 |   | 4 |
|   | 9 | 18 | 27 | 9–8 | 4–8 |   |   |
| 9 | 5 | 9 | 17 | 5–9 | 5–9 | 5 | 6 |
|   | 6 | 20 | 10 | 6–9 | 6–9 |   |   |
|   | 8 | 23 | 33 |   |   |   |   |
| 10 | 8 | 20 | 36 | 9–10 | 9–10 | 9 | 8 |
|   | 9 | 13 | 22 |   |   |   |   |

## Performance analysis

In this experiment, our primary goal is to select the optimal route that meets the needs of an application between nodes 1 and 2. The selection efficiency is assessed in the following three scenarios.

### Single and two routing metrics

Figure 5 depicts the optimal route determined by a single routing metric. Figure 5A illustrates the chosen route using hop count(traditional LOADng), Fig. 5B depicts the chosen route using the energy metric as demonstrated in our previous work (*Laouid et al., 2017*), Fig. 5C depicts the chosen route using the node cost, and Fig. 5D describes the

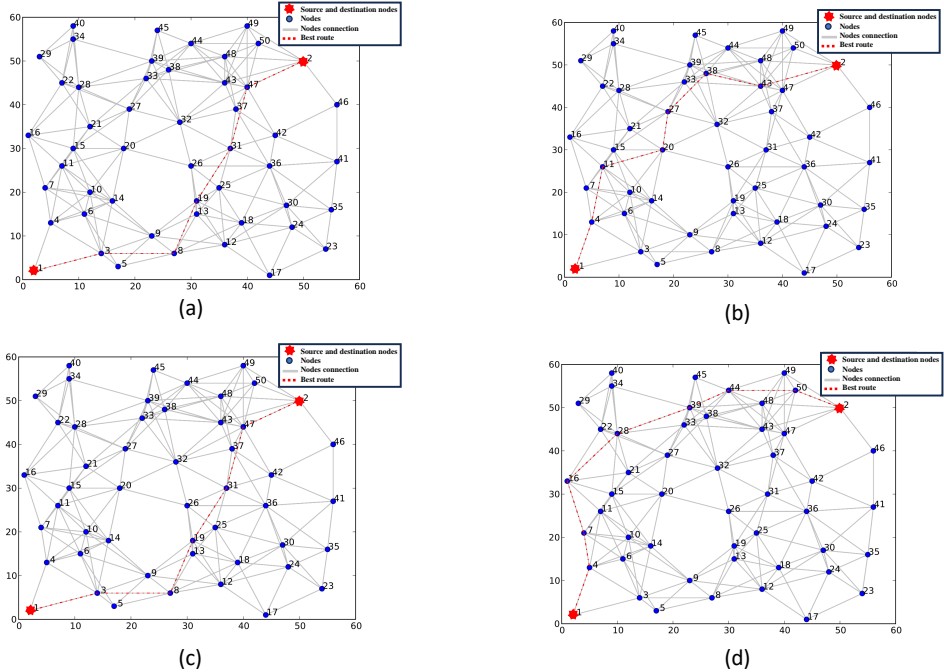

**Figure 5** (A–B) Selection of the best route using a single routing metric.

chosen route using the transmission delay. It is evident from the various figures that routes with a greater or smaller number of hops are chosen for different requests.

There may need to be more than a single measure to meet the needs of the applications. For example, using node cost as the only routing metric in selecting the optimal route might result in a higher transmission latency. Note that applications that use the same routing measure may require differing metric weights. Figure 6 shows the optimal route chosen using the node cost and the transmission latency routing metrics with varying weightings. Figure 6A shows the chosen route in a request with a weighted cost of 0.6 and a latency of 0.4. Figure 6B depicts the selected route in another request with a weighted cost of 0.7 and a delay of 0.3. Figures 6A and 6B show that different results are obtained for various requests with different weightings while using the same routing metric.

### Three and multiple routing metrics

Figure 7 shows the optimal route chosen using three routing parameters, namely, node cost, node energy, and transmission delay, with varying weightings. Figure 7A shows the preferred route in a request with weighted costs of 0.3, 0.2, and 0.5 respectively. Figure 7B depicts the chosen route in another request where the weighted costs are 0.2, 0.5, and 0.3. Figures 7A and 7B illustrate that when the same metric routing is utilized, we obtain different results for various requests (different weightings).

To show the scalability of P2P-IoMT in terms of the number of metrics, we re-executed the experiments using six metrics with different weights and an increased link density.

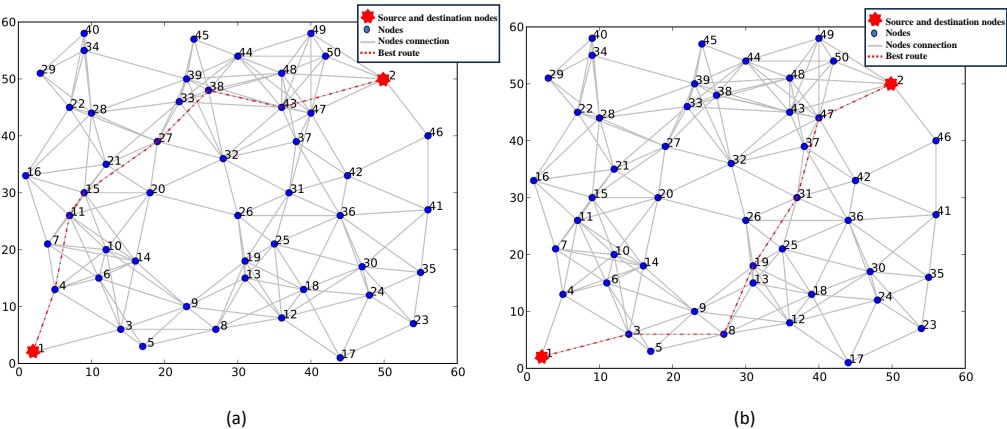

**Figure 6** (A–B) Selection of the best route using node cost and transmission delay as routing metrics.

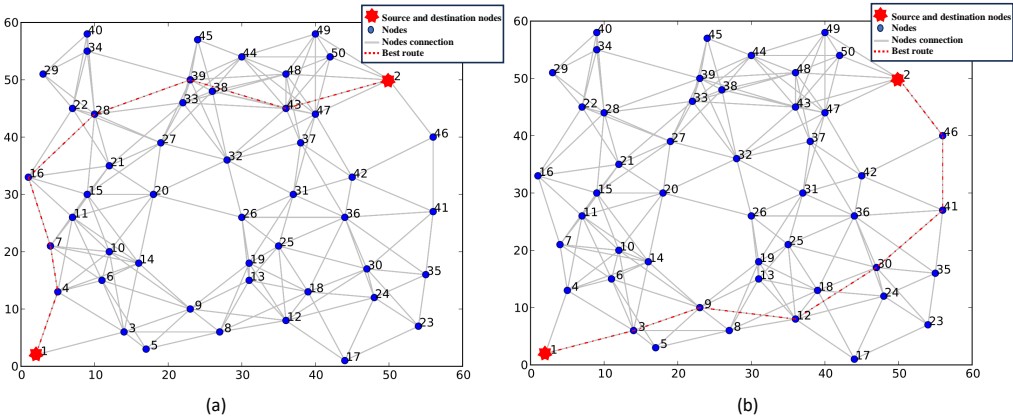

**Figure 7** (A–B) Selection of the best route using node cost, node energy, and transmission delay as routing metrics.

Figure 8 shows the optimal route with six routing metrics (energy, delay, service cost, link quality, security, and availability level) with two different requests.

### Complexity

The complexity of a routing protocol has a significant impact on its performance in terms of response time, energy consumption, and scalability (*Abuarqoub et al., 2012*). Measuring the complexity enables estimating the workload needed for data processing and routing decisions, optimizing efficiency, and prolonging node lifespan. Identifying potential network bottlenecks helps designers improve and optimize routing algorithms for reliable and predictable performance. Furthermore, complexity assessment helps to evaluate the feasibility and viability of routing techniques in resource-constrained environments, enabling adaptation of algorithms and strategies based on available resources. Quantifying routing complexity facilitates comparative studies, aiding researchers and practitioners in

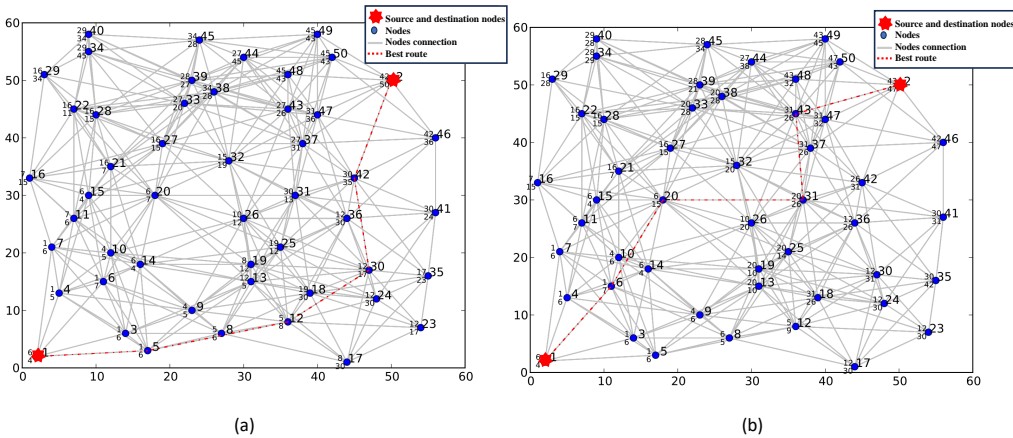

**Figure 8** **(A–B) Selection of the best route with six routing metrics.**

selecting the most suitable technique based on specific network objectives such as reliability, energy consumption, or latency.

The complexity formula to apply Skyline followed by Euclidean distance for the ranking also depends on the number of neighboring nodes ($n$) to evaluate and the number of routing metrics ($d$) used for object comparison. The overall complexity of the approach, applying Skyline first and then the Euclidean distance, can be represented as $O(n^2 * d) + O(n * log(n))$, where the first term represents the complexity of the Skyline calculation and the second term represents the complexity of ranking objects based on the Euclidean distance. The calculation of Skyline has a complexity of $O(n^2 * d)$, which is often more computationally expensive due to the need to compare each pair of objects to determine their dominance relationship. Once the Skyline is obtained, ranking the objects by Euclidean distance can be done using efficient data structures, reducing the complexity to $O(n * log(n))$ for the ranking step. It is important to note that this complexity formula is a general estimation, and improvements can be made by employing specific techniques, such as distance-based filtering, to reduce the number of objects evaluated by the Euclidean distance.

Due to the small number of immediate neighbors in IoMT environments, the complexity of the Skyline operator followed by the Euclidean distance decrease. The low complexity offers advantages such as reduced computation time, faster ranking, improved visualization, more efficient resource utilization, and increased accuracy.

## Performance comparison

In this section, we compare our P2P-IoMT's performance to LOADng (with hop count as the only routing measure) and LRRE (*Sasidharan & Jacob, 2018*). LRRE's composite routing metric reduces network congestion and extends nodes' life. Three routing metrics are used to select the best route, namely, hop count, residual energy, and the total number of live routes on a node. The same experimental environment was utilized, and the simulation was run five times with the average value used as the outcome. Simulation results show that P2P-IoMT performance exceeds its best rivals in the literature.

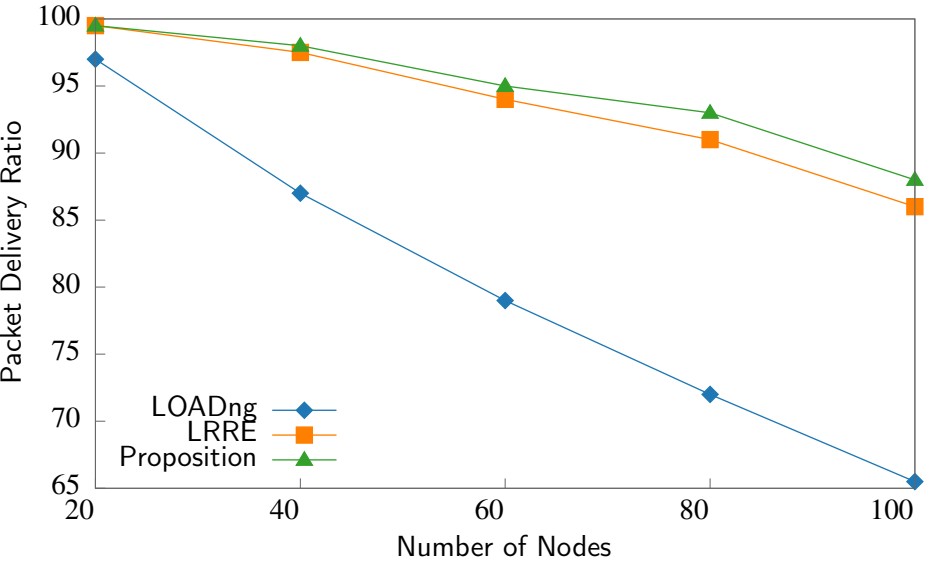

**Figure 9  PDR comparison.**

Four evaluation parameters are utilized to measure the efficiency of three compared routing strategies. The first parameter is the packet delivery ratio (PDR), which is a measure of reliability and is computed as follows (*Sasidharan & Jacob, 2018*):

$$PDR = 100 * \frac{number - of - packets - delivered}{number - of - packets - generated}$$

where the number of packets created equals the total number of packets generated by source nodes, and the number of packets delivered equals the total number of packets received by destination nodes. Figure 9 compares the PDR *versus* the number of nodes in the network. We observe that the proposed P2P-IoMT exhibits the best PDR performance while LOADng performs worst. P2P-IoMT effectively avoids congested nodes and nodes with low residual energy based on availability level, link quality, and energy metrics during the route selection. Consequently, P2p-IoMT significantly reduces packet loss caused by congestion and node failures. By maintaining a constant node density, we observe that the average hop count required to reach the destination increases proportionally with the expansion of network nodes. Consequently, the PDR decreases as the number of nodes in the network increases.

The second evaluation metric is the node's average residual energy, which reflects how well the routing protocol can spread the load across the network. Residual energy measures the network's lifetime and is used for making energy management decisions. The average residual energy in the nodes is calculated by adding the residual energy levels of all the nodes in the network and then dividing this sum by the total number of nodes. This makes it possible to determine the average energy remaining per node in the network. The average residual energy is computed as follows (*Sasidharan & Jacob, 2018*):

$$Average\ Residual\ Energy = \frac{\sum_{i=1}^{N} Eng(n_i)}{N}$$

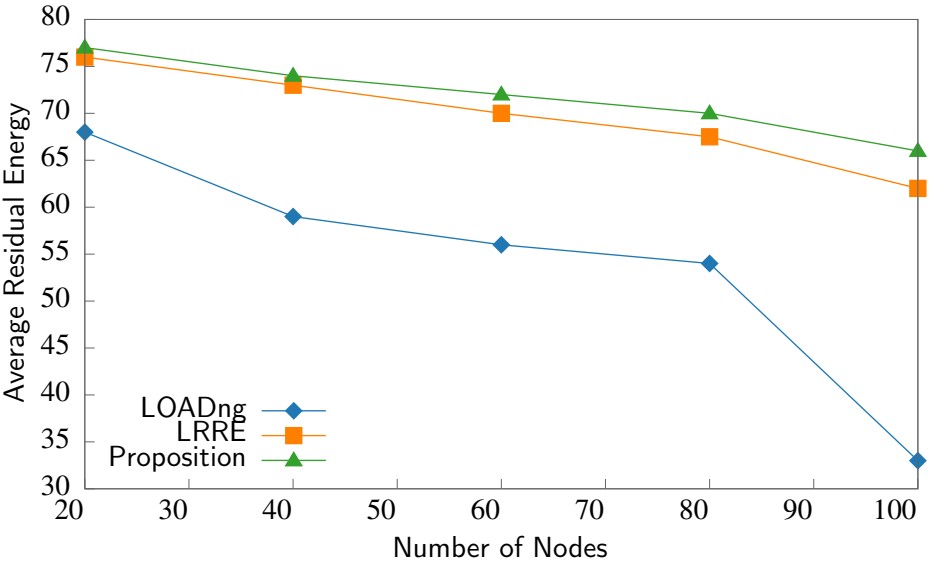

**Figure 10  Average residual energy comparison.**

where $Eng(n_i)$ is the remaining energy of the node $(n_i)$ (in percentage) and $N$ is the total number of nodes in the network. Figure 10 compares the network's average residual energy. Compared to LOADng and LRRE, P2P-IoMT exhibits the highest average residual energy, and the nodes have a longer lifetime, increasing the whole network's lifetime. Using the node energy level and link quality metrics in P2P-IoMT achieves better energy utilization. By favoring nodes with higher residual energy, the energy load in the network could be balanced and extend the overall network lifetime. The transmission quality between adjacent nodes when selecting routing paths is also considered. Prioritizing higher-quality links enhances data transmission efficiency and reduces energy-consuming re-transmission attempts. These factors prolong the network lifetime by balancing energy load among nodes, avoiding energy-depleted nodes, and congested routes.

The third factor is the network lifetime. The network lifetime refers to the period by which the network can operate before all nodes completely exhaust their power. This is a crucial measurement to assess the network's livability and energy efficiency. The network lifetime in IoMT is calculated using the energy capacity of the node's and the power consumption to perform different tasks. Figure 11 shows that P2P-IoMT has the longest lifetime compared with LOADng and LRRE. P2P-IoMT extends the network lifetime to 4% compared to the LRRE protocol.

The fourth performance evaluation metric is the path discovery time, which is critical in P2P routing. It represents the time required to find a valid path between $s$ and $d$. Several factors, such as the network size and topology, path discovery method, and the number of routing metrics, may influence the path discovery time of a routing protocol. In our implementation, the path discovery time varies from one experiment to another. Still, in

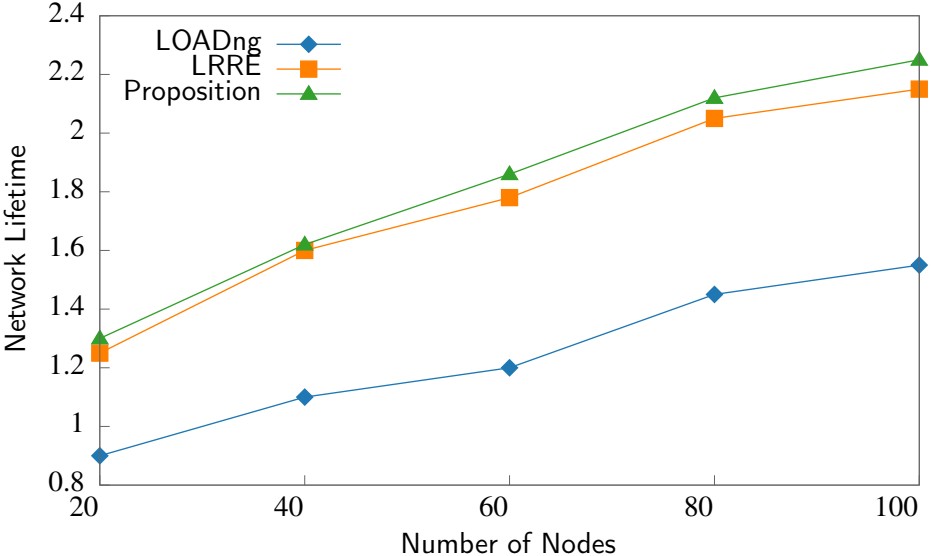

**Figure 11   Comparison of the network lifetime.**

general, P2P-IoMT is slightly lagging compared to other path discovery protocols for the following reasons:

- The number of metrics used increases the computation time
- Larger message size
- The two-phase (Skyline and Euclidean distance) path calculation consumes extra time

This slight delay is tolerable compared to the selected path quality, the increased average residual energy, and the long network lifetime.

In this study, we exploit the Skyline operator in multi-critical decision-making to choose the optimal paths. In contrast to the previous protocols, we have not defined the number of routing metrics or their weighting; every application can configure the relevant metrics and their weightings based on its specific needs. The many routing metrics applicable to IoMT are identified and defined.

## CONCLUSION

Efficient routing protocols are critical for the success of IoMT applications. Data routing becomes a challenging task considering IoT devices' resource limitations and the large network scale. This article presented a new P2P LOADng-based routing protocol that allows routes to be discovered dynamically on-demand. Nodes select the best parent when finding routes using dynamic composite routing metrics. The Skyline method is used in multi-criteria decision-making to determine the best route based on the application requirements. P2P-IoMT was evaluated in simulation, and the results showed that it significantly improved the PDR and network lifetime compared to its best rivals in the literature. While the P2P-IoMT protocol offers promising energy efficiency and packet delivery advantages, the complexity associated with Skyline can lead to increased path

discovery time, particularly as network size and the number of routing metrics grow. These scalability challenges reveal other concerns for mitigating them in our future research, specifically focusing on optimizing P2P-IoMT for diverse IoMT applications. In future work, P2P-IoMT will be compared against other recently published protocols. Moreover, the proposed composite routing metrics will be used as an objective function in constructing DODAG in the RPL protocol.

### Funding
The authors received no funding for this work.

### Competing Interests
The authors declare there are no competing interests.

### Author Contributions
- Ismail Kertiou conceived and designed the experiments, performed the computation work, prepared figures and/or tables, and approved the final draft.
- Abdelkader Laouid performed the experiments, prepared figures and/or tables, and approved the final draft.
- Benharzallah Saber analyzed the data, authored or reviewed drafts of the article, and approved the final draft.
- Mohammad Hammoudeh analyzed the data, authored or reviewed drafts of the article, and approved the final draft.
- Muath Alshaikh performed the experiments, authored or reviewed drafts of the article, and approved the final draft.

### Data Availability
The code and data are available in the Supplemental Files.

### Supplemental Information
Supplemental information for this article can be found online at http://dx.doi.org/10.7717/peerj-cs.1682#supplemental-information.

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
