# Peer review of "A P2P multi-path routing algorithm based on Skyline operator for data aggregation in IoMT environments"

_PeerJ Computer Science, doi:10.7717/peerj-cs.1682_

## Round 0.1 · original submission · Major Revisions

Note that to speed up the re-review process, please provide a detailed response letter. Please address each of the reviewers' points.
Thank you
C.S.

Reviewer 1 ·

Basic reporting

In this article, authors try to propose a new P2P LOADng-based routing protocol that allows routes to be discovered dynamically on-demand by leveraging the effectiveness of the
skyline technique.
The study seems to be interesting especially in IoMT applications based on the P2P communication. Unfortunately, several shortcomings have been observed throughout this paper. To further improve the content of this article, hereafter, some remarks and questions that should be taken into consideration :
- The article is quite well written. In my opinion, it would be better to revise the structure of the related work section. For instance, you might group all the articles focusing on the same issue together. Also, why the authors have cited research related to the security issue in IoMTs, which is a different issue to the one addressed in the article?
In the introduction section, you write "based on the limitations of current approaches", but you don't mention these limitations in the preceding paragraphs?
- The authors haven't properly explained why they have chosen Loadng P2P instead of RPL P2P.
- In page 3/19 line 126 :We're in 2023, so the limits invoked in the (Sousa et al. (2017); Sobral et al. (2019b)) references have not been taken into account by Contiki-NG?
- In the same page : why did you say "Unlike the RPL active method " while RPL is a proactive routing ?
- In section 2.1, the authors should cite references for each equation used to calculate each routing metric.
In equation (6), how the SL metric is calculated?
- What did you mean by the availability level?
- Figures 1(a), (b) and (c)  are misplaced, normally they should appear after the Route selection subsection.
- Review equations  (8) and (9), I think there is redundancy.

Some spelling and grammatical typos :
- all in the paper replace MIoT with IoMT
- Rephrase this sentence "......that each node in the routing table has several entries arranged ...." with "....that the routing table of each node has several entries arranged ...."
- This sentence "the different needed routing metrics of the objective function are less than a specified ...........to find other paths"  " is poorly expressed please rephrase it.
- Replace "ascribed" by "described".
- In algorithm 1, replace "BestR"  by "BestP".
- Table 4. List of abbreviations
- Check the line 244, I think that the ∀ symbol is extra.

Experimental design

- The operating principle of the Loadng-based routing protocol proposed by the authors isn't clear enough: you say that it's the gateway that makes the first broadcast, and each node builds its own routing table containing the paths to the gateway (table 2), so this table can be used to route tarffic of a MP2P-type application to the gateway, so where is the P2P communication?
-The multi-path technique is not highlighted in the authors' proposal. does this mean that for each node several paths are discovered, or what exactly?
- Page (7/19) : the RMi field is not well explained, what it contains exactly?
- The Dest Adr field of the P2P-RReq contains normally the broadcast address since the P2P-RReq is a broadcast packet.
- A flow chart to explain how the proposed routing protocol works might be preferable.
- How to choose the  predetermined delay time required for waiting all the P2P-RReq?
- What did you mean by "the join route problem"?
- How you have declared and implemented the P2P-RReq packet whose size varies according to the needs of each application?
- In sub-section 3.3 Use case, in line 381 : before explaining this step, it is necessary to first explain the triggering of the routes discovery step using  P2P-RReq messages, which node triggers the discovery process? 
- In table 5, why there is not the routing table of node 1?
- The proposed routing protocol is based on the Skyline technique, so it is very important to assess the complexity of this proposal.

Validity of the findings

- The comparative study requires the evaluation of other performance metrics mainly related to P2P communication  such as the path discovery time, which certainly has an impact on real-time applications, and calculating the overhead in terms of the number of P2P-RREq messages that are broadcast during path discovery in order to build short routing paths and the length of the P2P discovered routes.
- In the Evaluation results and analysis section :  a brief overview on LRRE must be given, the routing protocol used to make a comparative study, or at least provide a reference for the interested readers.
- The authors did not specify the duration of the simulation, an important parameter that allowed us to test the performance of the proposal over a fairly significant period of time.
- Give a definition of the two other evaluation parameters in the same way as you did for PDR.

Therefore, it is recommended to authors to properly highlight their work and improve the proposal evaluation with more experiments in order to make it more suitable for publication.

Reviewer 2 ·

Basic reporting

The paper presents an improved routing protocol for IoMT that aims to overcome the limitations of current solutions in P2P data traffic scenarios. The proposed approach introduces adaptations on the LOADng, a well-studied routing protocol for wireless low power networks. The improvements go through the proposal of new routing metrics to be used accordingly to the applications and the use of the skyline technique to assist the nodes to select the best parents.
The aim of the proposal is well-justified and clearly presented.
The paper presents good references, and the background is enough. The structure of the text is as expected. However, the presented results are weak due to the limitation of the simulated scenario.

Experimental design

The problem studied in the paper is relevant but not new and is already well-studied in the literature. The proposed solution could not be considered innovative since the mixing of routing metrics to select the best path between two nodes is very common.
The related works session is weak and could better explore the limitations of studied solutions, one by one. The authors limit themselves to simply describing the functioning of related proposals in a few words and make general comments about the found drawbacks.
In sections 2 and 3, the authors should better use images, algorithms, and flowcharts to explain how the proposal works and how the nodes process each kind of control message during the route creation process. In general, the authors only describe the P2P-Rreq message not explaining the other control messages presumably used by the protocol as P2P-Rrep or messages for ACK, which are very common in reactive routing protocols. I suggest the authors dedicate some words to explain each type of control message used.
In section 2.2, as indicated previously, I suggest the author create an algorithm or flowchart to better explain the procedure made by the node when receives a P2P-RReq. It could help the readers to understand the route creation process.
In section 4 the authors present the results of the performance evaluation conducted based on simulations. It used only one network scenario, which is very little to ensure that the proposed solution is better than the others. Also, according to the authors, the data messages are always exchanged among nodes 1 and 2. This scenario cannot represent a real IoMT scenario and make conducted experiment weak and poorly reliable. I suggest the authors expand the study to other scenarios considering the performance of the proposal with different numbers of nodes, network deployments, and several nodes sending data messages to the others. It could help to observe the scalability (which is a key aspect in IoT networks) of the proposed solution.

Validity of the findings

Three performance metrics were observed: PDR, average residual energy, and network lifetime. Although widely used and important, these metrics are not enough to define an efficient routing protocol. Also, many comments about the results are limited to describing what can be seen in the graphics. The authors should better explain what lead the proposal to that better results in comparison to other approaches. Also, many comments about the results are limited to describing what can be seen in the graphics. The authors should better explain what lead the proposal to those better results in comparison to other approaches. Furthermore, are not provided the number of simulations executed to reach the results nor statistical data (e. g. error bars). It makes unreliable the presented performance results.
I suggest the authors consider other metrics such as the number of control messages, the amount of memory required by devices to execute the protocol and network functioning, end to end delay of data messages. This last one is very relevant to the type of application where the proposed protocol should be used, medical and health applications. Considering that reactive routing protocol should perform a route creation process before the sending of data packets, what is the impact of this creating process on the delay in the delivery of the data messages?
Considering this last point, the authors could justify why opting for a reactive protocol, which can suffer from high delays due to the need of creating routing on demand, for IoMT applications instead of a proactive protocol, which previously create the routes among the main nodes of the network.
Finally, considering that the authors are looking to the development of an improved P2P routing solution for IoMT, I suggest thinking about creating a simulated network scenario that better reflects this kind of application. The conducted performance study considered the sending of multi-hop data messages between only two nodes. Real application scenarios can easily require the exchange of messages among several network nodes which makes hardest the tasks of route creation and maintenance.

Additional comments

I suggest the use of proof reading services to ensure that paper text is free of language errors.

Also, some points can be fixed:
- Line 62, “goal functions” should be fixed to “objective functions”
- Line 143, LOADing should be fixed to LOADng
- Line 145, LOADing should be fixed to LOADng

Reviewer 3 ·

Basic reporting

In this article, the authors present a multi-path reactive point-to-point routing protocol based on LOADng and specifically designed for applications in the context of the Internet of Medical Things. The algorithm is dynamic and on-demand, namely when a node needs to send a message, it first sends a broadcast request message to its neighbors which initiates the route finding and selection process. The article introduces multiple routing metrics to describe a variety of IoT application constraints that are then used as a costs, based on which the best route is selected. The algorithm leverages the “Skyline” technique, a mechanism for decision-making in multi-criteria scenarios, to choose the best parent node.

The article is very well written, the use of English is correct, professional, and the sentences are clear and easy to read. A few minor changes and corrections are recommended in “Additional Comments”.

The article has an extensive literature review section (Related Work) with many recent publications that provide enough background to the study presented. However, stressing the limitations of each related work, or at least of the most important ones, would improve the quality and contribution of the article. In fact, the limitations are briefly summarized only at the end of the section (from line 210). Furthermore, expanding on the consequences of these limitations and how they would impact the network operations/performance in the IoMT framework would also provide a more understandable context for the reader.
In the discussion of the related work (lines 100 to 109) it is confusing which topic the article is discussing: if security, routing, or both.

A major weakness of the article is the presentation and organization of the first two sections ("Introduction" and "Related Work"). In some cases, there is no connection between one paragraph and the next one, and some concepts are introduced abruptly. As for example, from line 52 to 61, where the definition of wireless sensor network (WSN), although useful, is not clear. The communication capability in a WSN, the multi-hop strategy, and so on, are confusing to non-experts. The two scenarios described from line 56 can provide examples of specific cases of IoMT but it is not clear which concept they are trying to support. Moreover, they are not used later in the paper, for example as simulation scenarios, as the reader might expect.
Then, on line 61, the IPv6 is abruptly introduced and the Introduction, from presenting a general overview of the problem, becomes very technical, and the protocols reported in the next paragraphs and how they relate to each other are not clear. Consequently, also the following motivations (lines 63-75) for this work are not strong enough. For these reasons I suggest revising the first two sections of the article.
In addition, it would be helpful to provide a definition of Sensing as a Service (SaaS), how it relates to IoT, WSNs and the proposed solution.

Experimental design

It can be appreciated how the article reports on the design choices (e.g., point-to-point approach rather than a different one), but it should also further explain the reasons why these choices are preferable to other existing approaches.

Although placing this work in the context of the IoT and WSNs is appropriate, specifying the proposed algorithm for the Internet of Medical Things (IoMT) requires further explanation. The authors might want to provide stronger argument(s) on why this work is a good fit for the IoMT, what constraints it derives form it, which issues it addresses that are specific to the IoMT, and how it tries to solve them. In fact, in the final evaluation (Section 4), “50 randomly distributed nodes throughout a 60×60m^2 space” are considered, which is acceptable in a generic IoT/WSN scenario but seems unlikely (too many nodes) in an IoMT scenario or a Wireless Body Area Networks (WBAN).

Finally, the metrics adopted to optimize the protocol (Section 2) seem to be relevant for the cases the protocol is designed for, but a brief discussion on why each one specifically is selected and important in the IoMT should be added.

Validity of the findings

From a technical perspective, here are a few questions I am left with by reading the article: how is the proposed approach different from other routing protocols that consider multiple metrics too? What are the limitations or drawback of this protocol (e.g., increased computational complexity, processing time/energy)? How do the complexity and energy consumption increase with the number of metrics used?

The experimental evaluation provides good data to understand how the protocol works, its basic functions, and how it performs compared to the state of the art. In addition to these results, it would be interesting to have more data on how the performances change with increasing the number of metrics (even though this could be challenging to compare to existing protocols). In fact, this is a strength of the proposed solution, and having results for only 2 or 3 metrics (which is what many other routing protocols can do) almost defeats the purpose of this protocol.
In the context of the IoMT it is relevant also to show results for a network with a small number of nodes (<20).

In Section 4.2, besides the final route itself, what are the total costs of the final route in terms of the two or three metrics selected, i.e. hop count, nodes energy, nodes cost, and transmission latency.?

Additional comments

Here are a few minor comments:

The last sentence of the abstract is too generic, some numerical results of the efficiency or improved performance of the proposed method could be added.

Please be consistent with acronyms. IoMT often written as MIoT (lines 79, 129, 196)

Line 94: “sensor nodes implanted on a human body”, should be implanted “in” or “attached to” a human body, or even both for a WBAN.

Acronyms OFs (line 205), LQI (line 199), ETX (238) used for the first time and are not defined.

“and these exists k with” at equation (8), does not read well, please revise.

The messages on lines 136-139 can be removed because they are not used in the article or to explain how the algorithm works.

Figures 7, 8, and 9 can be smaller.


Authors, I'm looking forward to read your revised version.

---

## Round 0.2 · Minor Revisions

Please provide the revision according to the reviewers' comments.

Reviewer 1 ·

Basic reporting

In general, the answers provided by the authors are globally convincing. Nevertheless, here are a few simple points to take into account :

- In the abstract, the authors forgot to mention that P2P-IoMT is the name of their proposal.

- From line 58 to 68 : The authors introduced RPL and LOADng, then from 69 to 75, they moved abruptly on to multipath P2P routing protocols, citing different examples and limitations, then went on to cite their contributions without specifying that their P2P-IoMT is a multipath routing protocol. I suggest that the authors revise the passage between these paragraphs.

- About the evaluation of the overhead performance metric, it may be that the number of P2P-RREq messages used by P2P-IoMT is identical to that of AODV, but the overhead related to the size of these messages is different for these two routing protocols.

- In the added Complexity section, it must be specified what the objects (n) and dimension (d) represent in the context of the proposed routing protocol (number of neighbouring nodes, number of routing metrics, etc.). It is clear that Skyline's complexity increases with network size and density, and also with the number of routing metrics to be considered for a given IoMT application. This leads to an increase in path discovery time, and consequently P2P-IoMT scalability remains a challenge.
So, in the conclusion section, it's important to point out the limitations of the proposed routing protocol, in particular the path discovery time, and clarify that these limitations will be taken into account in the future work, particularly with regard to IoMT applications.

- Line 63, LOADG should be fixed to LOADng
- Line 91 : Section 6 should be fixed to Section 5
- Line 93 : I suggest to add applications at the end of “Given the strict QoS requirements of IoMT ”.
- line 404: to all their neighbouring nodes should be fixed to to all its neighbouring nodes

Experimental design

No comment

Validity of the findings

No comment

Additional comments

No comment

Reviewer 3 ·

Basic reporting

The authors have addressed the comments and the concerns of the reviewers and have provided a major revision of the manuscript. The article is well written, the use of English is correct. The presentation has improved and especially the first two sections are better structured and provide the details necessary to position the article within the IoMT context and understand the proposed solution.

The authors improved the literature references reporting the limitations of the related works as recommended by the reviewers.

Experimental design

The experimental design section was improved by adding more scenarios and simulation results.

Validity of the findings

No comment

Additional comments

Please replace the word “determinantal” on line 42 which refers to a mathematical concept

---

## Round 0.3 · accepted · Accept

Based on the feedback received, the paper now is ready for publication, thank you for your consideration in PeerJ Computer Science.
Thank you
C.S.